# Negative regulation of urokinase receptor activity by a GPI-specific phospholipase C in breast cancer cells

**Michiel van Veen[1†], Elisa Matas-Rico[1†], Koen van de Wetering[2‡], Daniela Leyton-Puig[1], Katarzyna M Kedziora[1§], Valentina De Lorenzi[3], Yvette Stijf-Bultsma[4], Bram van den Broek[1], Kees Jalink[1], Nicolai Sidenius[3], Anastassis Perrakis[4], Wouter H Moolenaar[1*]**

[1]Division of Cell Biology, The Netherlands Cancer Institute, Amsterdam, Netherlands; [2]Division of Molecular Oncology, The Netherlands Cancer Institute, Amsterdam, Netherlands; [3]IFOM, The FIRC Institute of Molecular Oncology, Milan, Italy; [4]Division of Biochemistry, The Netherlands Cancer Institute, Amsterdam, Netherlands

**\*For correspondence:**
w.moolenaar@nki.nl

[†]These authors contributed equally to this work

**Present address:** [‡]Department of Dermatology and Cutaneous Biology, Sidney Kimmel Medical College at Thomas Jefferson University, Philadelphia, United States; [§]University of North Carolina, Chapel Hill, United States

**Competing interests:** The authors declare that no competing interests exist.

**Abstract** The urokinase receptor (uPAR) is a glycosylphosphatidylinositol (GPI)-anchored protein that promotes tissue remodeling, tumor cell adhesion, migration and invasion. uPAR mediates degradation of the extracellular matrix through protease recruitment and enhances cell adhesion, migration and signaling through vitronectin binding and interactions with integrins. Full-length uPAR is released from the cell surface, but the mechanism and significance of uPAR shedding remain obscure. Here we identify transmembrane glycerophosphodiesterase GDE3 as a GPI-specific phospholipase C that cleaves and releases uPAR with consequent loss of function, whereas its homologue GDE2 fails to attack uPAR. GDE3 overexpression depletes uPAR from distinct basolateral membrane domains in breast cancer cells, resulting in a less transformed phenotype, it slows tumor growth in a xenograft model and correlates with prolonged survival in patients. Our results establish GDE3 as a negative regulator of the uPAR signaling network and, furthermore, highlight GPI-anchor hydrolysis as a cell-intrinsic mechanism to alter cell behavior.
DOI: https://doi.org/10.7554/eLife.23649.001

## Introduction

The urokinase-type plasminogen activator receptor (uPAR) is a central player in a complex signaling network implicated in a variety of remodeling processes, both physiological and pathological, ranging from embryo implantation to wound healing and tumor progression (*Boonstra et al., 2011*; *Ferraris and Sidenius, 2013*; *Smith and Marshall, 2010*). uPAR is a glycosylphosphatidylinositol (GPI)-anchored protein and hence lacks intrinsic signaling capacity. Instead, uPAR acts by binding two major ligands, namely the protease urokinase plasminogen activator (uPA) and the extracellular matrix (ECM) protein vitronectin (*Ferraris and Sidenius, 2013*; *Madsen et al., 2007*; *Smith and Marshall, 2010*). Through uPA binding, uPAR localizes plasmin generation to the cell surface and thereby promotes pericellular proteolysis and ECM degradation (*Ferraris and Sidenius, 2013*; *Smith and Marshall, 2010*). In addition, through vitronectin binding and functional interactions with integrins and growth factor receptors, uPAR activates intracellular signaling pathways leading to cytoskeletal reorganization, enhanced cell adhesion and motility and other features of tissue remodeling and cell transformation (*Ferraris et al., 2014*; *Madsen et al., 2007*; *Smith and Marshall, 2010*). As such, uPAR is a master regulator of extracellular proteolysis, cell motility and invasion. uPAR expression is elevated during inflammation and in many human cancers, where it often

**eLife digest** Every process in the body, from how cells divide to how they move around, is tightly regulated. For example, cells only migrate when they receive the correct signals from their environment. These signals are recognised by receptor proteins that sit on the cell surface and connect the outside signal with the cell's response. However, in cancer cells, these processes are out of control, which is why cancer cells can grow very quickly or spread to many different parts of the body.

One important receptor protein is the urokinase receptor, which helps to reorganize the tissue, for example, when wounds heal, but also enables cancer cells to grow and spread. A special feature of urokinase receptor is the way it is connected to the cell surface, namely through a molecule that acts as an anchor, called the GPI anchor. The urokinase receptor and some other GPI-anchored proteins can be released from their anchor. However, until now it was not clear why and how the urokinase receptor is released from cells, or how losing the receptor affects the cell.

Now, van Veen, Matas-Rico et al. studied breast cancer cells, and discovered that an enzyme called GDE3 cuts the urokinase receptor off its GPI anchor to release the receptor from the cells. However, when breast cancer cells shed the urokinase receptor, they also lost the receptor from the cell surface in specific areas. As a result, the receptor could not work anymore. When breast cancer cells were experimentally modified to produce high levels of GDE3, the cancer cells became less mobile and aggressive.

Van Veen, Matas-Rico et al. then implanted 'normal' breast cancer cells, and breast cancer cells with extra GDE3 into mice, and observed that the tumors of mice with additional GDE3 grew less quickly. Moreover, breast cancer patients with high levels of GDE3 tend to live longer than patients with low levels of GDE3. These results suggest that the enzyme GDE3 can suppress tumor growth.

These findings uncover a new way how cells can alter their behavior, namely by cleaving GPI anchors at the cell surface. Future experiments will need to address how GDE3 itself is controlled, and if it releases other GPI-anchored proteins from cells. Once we know how to increase GDE3 activity in tumor cells, the new knowledge could one day lead to therapies to help patients with cancer.

DOI: https://doi.org/10.7554/eLife.23649.002

correlates with poor prognosis, supporting the view that tumor cells hijack the uPAR signaling system to enhance malignancy (*Boonstra et al., 2011*; *Ferraris and Sidenius, 2013*; *Smith and Marshall, 2010*). Increased uPAR expression in solid tumors and the corresponding activated stroma is being evaluated by PET-imaging for patient stratification (*Persson et al., 2015*).

It has long been known that full-length uPAR is released from the plasma membrane resulting in a soluble form (suPAR) (*Pedersen et al., 1993*; *Ploug et al., 1992*), which is detectable in body fluids and considered a marker of disease severity in cancer and other life-threatening disorders (*Haupt et al., 2012*; *Hayek et al., 2016*; *Shariat et al., 2007*; *Sidenius et al., 2000*; *Stephens et al., 1999*). Circulating suPAR is derived from activated immune and inflammatory cells (*Ferraris and Sidenius, 2013*; *Smith and Marshall, 2010*), and also from circulating tumor cells (*Mustjoki et al., 2000*).

Locally produced suPAR might function as a ligand scavenger to confer negative feedback on uPAR (*Smith and Marshall, 2010*). In addition, both uPAR and suPAR can undergo proteolytic fragmentation by uPA and other proteases, possibly leading to new signaling activities (*Montuori and Ragno, 2009*). Yet, despite decades of research, the mechanism of uPAR release and its physiological implications have been elusive. A GPI-specific phospholipase D (GPI-PLD) (*Scallon et al., 1991*) has often been assumed to mediate the shedding of GPI-anchored proteins, but this unique PLD does not function on native membranes (*Mann et al., 2004*).

A possible clue to the mechanism of uPAR release comes from recent studies showing that a member of the glycerophosphodiester phosphodiesterase (GDPD/GDE) family (*Corda et al., 2014*), termed GDE2, promotes neuronal differentiation by cleaving select GPI-anchored proteins, notably a Notch ligand regulator and heparan sulfate proteoglycans (glypicans) (*Matas-Rico et al., 2016*; *Matas-Rico et al., 2017*; *Park et al., 2013*). GDE2, along with GDE3 and GDE6, belongs to a GDE

subfamily characterized by six-transmembrane-domain proteins with a conserved catalytic ectodomain (*Figure 1A*) (*Corda et al., 2009*; *Matas-Rico et al., 2016*). GDE2's close relative, GDE3, accelerates osteoblast differentiation through an unidentified mechanism (*Corda et al., 2009*; *Yanaka et al., 2003*), while the function of GDE6 is unknown.

Here we identify GDE3 as the first mammalian GPI-specific phospholipase C (GPI-PLC) that cleaves and sheds uPAR with consequent loss of uPAR activities in both HEK293 and breast cancer cells.

## Results

### GDE3, but not GDE2, sheds uPAR from HEK293 cells

We set out to determine whether uPAR can be released by any of the three related GDE family members, GDE2, GDE3 and GDE6. When expressed at relatively low levels in HEK293 cells, human GDE2 and GDE3 (HA-tagged) localized to distinct microdomains at the plasma membrane, possibly representing clustered lipid rafts where GPI-anchored proteins normally reside (*Maeda and Kinoshita, 2011*), as well as to filopodia-like extensions (*Matas-Rico et al., 2016*) (*Figure 1—figure supplement 1A,B*). By contrast, human GDE6 was mainly detected in intracellular compartments and therefore was not further tested (*Figure 1—figure supplement 1A*).

To assess GDE activity, we generated stable uPAR-expressing HEK293 cells (HEK-uPAR cells), expressed GDE2 and GDE3 and examined the appearance of suPAR in the medium, using bacterial phospholipase C (PI-PLC) as a positive control (*Matas-Rico et al., 2016*). Strikingly, uPAR was readily released into the medium by GDE3 and PI-PLC, but not by GDE2 (*Figure 1B*). GDE3 competed with exogenous PI-PLC to deplete uPAR, since PI-PLC was much less efficient in GDE3-overexpressing than in control HEK-uPAR cells (*Figure 1—figure supplement 1C*).

Mutating putative active-site residue H229, corresponding to H233 in GDE2 (*Matas-Rico et al., 2016*), abolished GDE3 activity without affecting its membrane localization (*Figure 1C*; *Figure 1—figure supplement 1D*). Furthermore, a transmembrane version of uPAR (uPAR-TM) lacking the GPI moiety (*Cunningham et al., 2003*) was resistant to GDE3 attack, consistent with GDE3 acting through GPI-anchor hydrolysis (*Figure 1—figure supplement 1E*). Flow cytometry analysis of GDE3-overexpressing cells confirmed decreased uPAR levels in the plasma membrane (*Figure 1D*), while TIRF microscopy revealed substantial uPAR loss from the ventral cell surface (*Figure 1E*).

### GDE2 versus GDE3: homology modeling

The selectivity of GDE3 versus GDE2 towards uPAR cleavage is striking. In an attempt to understand the structural basis of this selectivity, we constructed homology-based models of the globular α/β barrel GDPD domains, using I-TASSER (*Yang et al., 2015*). We reasoned that the catalytic domains must recognize not only the PI lipid moiety at the membrane-water interface, but also the attached protein. While GDE2 and GDE3 have a similar putative GPI-binding groove leading to the active site, they show striking differences in their surface charge distribution, particularly at the putative substrate interaction surface (*Figure 1F*). It therefore seems likely that surface properties are a major determinant of selective substrate recognition by GDEs, a notion that should be validated by future structural studies.

### GDE3 is a GPI-specific phospholipase C that cleaves uPAR in cis

We asked whether GDE3 attacks uPAR in cis (same cell) or in trans (adjacent cell), or both. By mixing GDE3-expressing cells (lacking uPAR) with uPAR-expressing cells (lacking GDE3), GDE3-expressing cells failed to shed uPAR from the GDE3-deficient cell population (*Figure 2A*). Thus, GDE3 acts in cis, attacking uPAR on the same plasma membrane, not on adjacent cells. To determine whether GDE3 acts as a phospholipase in GPI-anchor cleavage, we used Triton X-114 partitioning and liquid chromatography-mass spectrometry (LC-MS). Triton X-114 partitioning revealed that suPAR did not contain lipid moieties, as it was not detected in the detergent phase (*Figure 2B*). Next, suPAR was immunoprecipitated from the medium of GDE3-expressing cells and treated with nitrous acid (HONO) to cleave the glucosamine-inositol linkage in the GPI core (*Figure 2C*). Subsequent analysis by LC-MS revealed that the acid-treated suPAR samples contained inositol 1-phosphate (*Figure 2D*). This result defines GDE3 as the first mammalian GPI-specific phospholipase C (GPI-PLC).

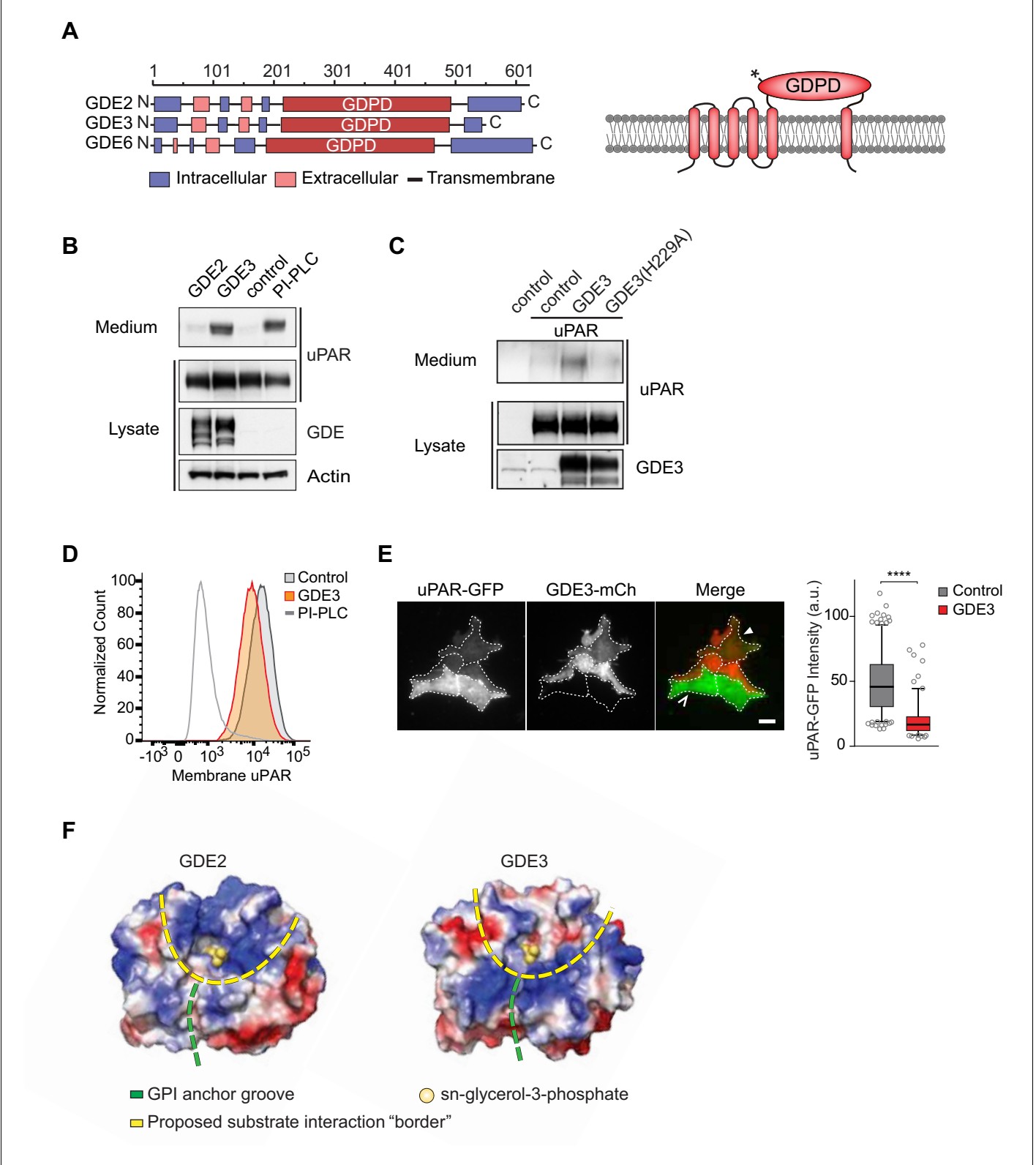

**Figure 1.** GDE3, but not GDE2, sheds uPAR from HEK293 cells. (**A**) Domain structure of GDE2, GDE3 and GDE6 (left panel), and the transmembrane scheme of GDE3 (right panel). GDPD denotes catalytic glycerophosphodiesterase domain. Asterisk in GDPD domain depicts catalytic His residue in both GDE2 and GDE3. (**B**) Immunoblot analysis of uPAR release into the medium. HEK-uPAR cells transfected with empty vector (control), GDE2 or GDE3. PI-PLC served as positive control. (**C**) Mutant GDE3(H229A) fails to release uPAR. (**D**) Partial loss of uPAR from the plasma membrane by GDE3,

*Figure 1 continued on next page*

Figure 1 continued

as measured by flow cytometry. (E) TIRF microscopy reveals loss of uPAR from the basolateral plasma membrane. Box plot shows uPAR-GFP intensity at the ventral membrane (n = 3, mean ±SEM ****p<0001). (F) Homology modeling of the GDE2 and GDE3 catalytic domains showing surface charge distributions (blue, positive; red, negative; green line, putative GPI-binding groove; yellow line, proposed substrate-binding surface). The active site is indicated by glycerol-3-phosphate located at the template structure.

DOI: https://doi.org/10.7554/eLife.23649.003

The following figure supplement is available for figure 1:

Figure supplement 1. GDE subcellular localization and induction of uPAR release from HEK293-uPAR cells.

DOI: https://doi.org/10.7554/eLife.23649.004

## GDE3 suppresses the vitronectin-dependent activities of uPAR

When compared to uPAR-deficient cells, HEK-uPAR cells showed markedly increased cell adhesion, loss of intercellular contacts and enhanced spreading with prominent lamellipodia formation on vitronectin, but not on fibronectin (*Figure 3A*), typical features of a Rac-driven motile phenotype, in agreement with previous studies (*Kjøller and Hall, 2001*; *Madsen et al., 2007*). Cell spreading coincided with activation of focal adhesion kinase (FAK), indicative of integrin activation (*Figure 3B*). Strikingly, expression of GDE3 largely abolished the uPAR-induced phenotypes and cellular responses (*Figure 3A–F*). Catalytically dead GDE3(H229A) had no effect, neither had overexpressed GDE2 (*Figure 3F* and results not shown). Thus, by releasing uPAR from the plasma membrane, GDE3 suppresses the vitronectin-dependent activities of uPAR. Treatment of diverse cell types with suPAR-enriched conditioned media from HEK293 cells did not evoke detectable cellular responses, supporting the notion that suPAR is biologically inactive, at least under cell culture conditions.

## GDE3 locally depletes uPAR and suppresses its activities in breast cancer cells

We next assessed the impact of GDE3 on endogenous uPAR activity in MDA-MB-231 **t**riple-negative breast cancer cells. These cells express relatively high levels of uPAR (*Figure 4A*) and its ligand uPA (*LeBeau et al., 2013*), thus forming an autocrine signaling loop. Expression of GDE3 (encoded by *GDPD2*) is relatively low in breast cancer lines (n = 51), including MDA-MB-231 cells (*Barretina et al., 2012*) (*Figure 4B*). GDE3 expression in MDA-MB-231 cells led to a modest loss of uPAR from the plasma membrane as shown by flow cytometry (*Figure 4C*). To determine how GDE3 expression affects localized uPAR levels at the basolateral plasma membrane, we used confocal and dual-color super-resolution microscopy in TIRF mode. Strikingly, basolateral membrane microdomains containing wild-type GDE3 showed little or no colocalization with endogenous uPAR. In marked contrast, catalytically dead GDE3(H229A) clearly colocalized with uPAR in those membrane domains (*Figure 4D*). Quantification of co-localization was performed on multiple cells (*Figure 4D*, right panel). These results strongly suggest that active GDE3 depletes uPAR levels from distinct domains at the basolateral membrane. Consistent with this, CRISPR-based knockout of GDE3 resulted in increased basolateral uPAR levels when the cells were plated on vitronectin (*Figure 4E,F*; *Figure 4—figure supplement 1*).

Wild-type MDA-MB-231 cells adopted a motile phenotype on vitronectin, as evidenced by increased cell spreading with marked lamellipodia formation (*Figure 5A–C*), strongly reminiscent of a uPAR-regulated phenotype. Overexpressed GDE3 abolished the vitronectin-dependent phenotype of MBD-MB-231 cells (*Figure 5A–C*). Very similar effects of GDE3 overexpression were observed in another uPAR-positive breast cancer cell line (triple-negative Hs578T cells) (*Figure 5—figure supplement 1*). Of note, no effects were observed upon GDE2 overexpression in these cells (data not shown).

To confirm that GDE3 acts through uPAR attack, we expressed non-cleavable uPAR-TM to compete out endogenous uPAR, and found that the GDE3-induced reduction of cell spreading on vitronectin was largely inhibited (*Figure 5B*). Furthermore, shRNA-mediated knockdown uPAR (*Figure 5D*) gave rise to the same phenotype as GDE3 overexpression, namely reduced cell adhesion, spreading and lamellipodia formation on vitronectin (*Figure 5E,F,G*).

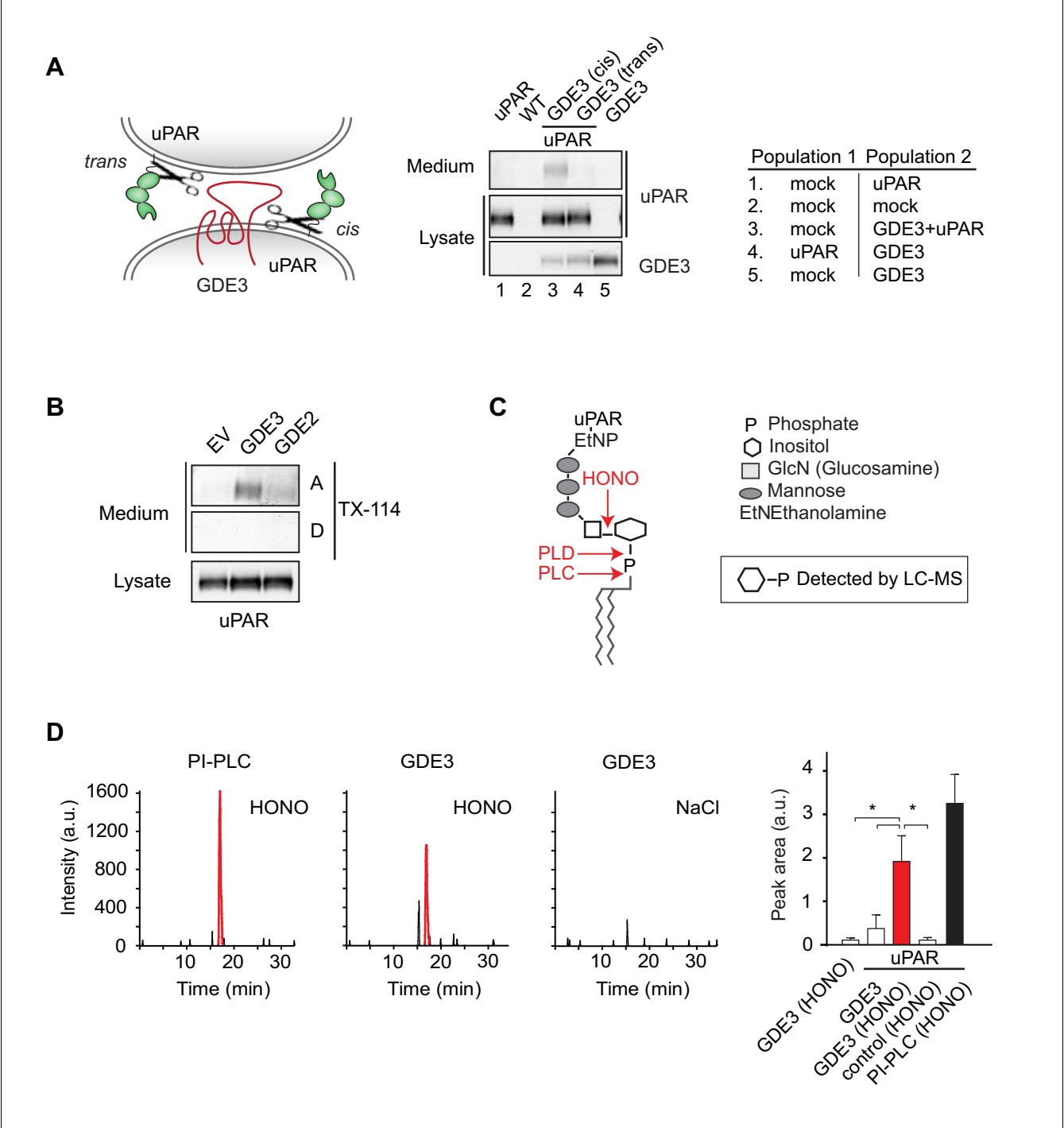

**Figure 2.** GDE3 is a GPI-specific PLC that cleaves uPAR in cis. (**A**) (Left) Scheme showing uPAR cleavage in cis or trans. (Right) GDE3-expressing HEK-uPAR cells were mixed with a GDE3-deficient cell population, as indicated. Immunoblot analysis of uPAR in medium and cell lysates indicates that GDE3 acts in cis; mock refers to empty vector-transfected cells. (**B**) GDE3 expression leads to increased GPI-free suPAR. Conditioned medium from HEK-uPAR cells expressing GDE2 or GDE3 was subjected to Triton X-114 phase separation. suPAR in the aqueous (**A**) and detergent (**D**) fractions was analyzed by immunoblotting. (**C**) GPI-anchor with phospholipase cleavage sites indicated; HONO, nitrous acid. (**D**) Representative LC-MS ion chromatograms (m/z 259.02–259.03); inositol 1-phosphate peaks in red. HONO-treated suPAR contains inositol 1-phosphate (n = 3, mean ±SEM; *p<0.05).
DOI: https://doi.org/10.7554/eLife.23649.005

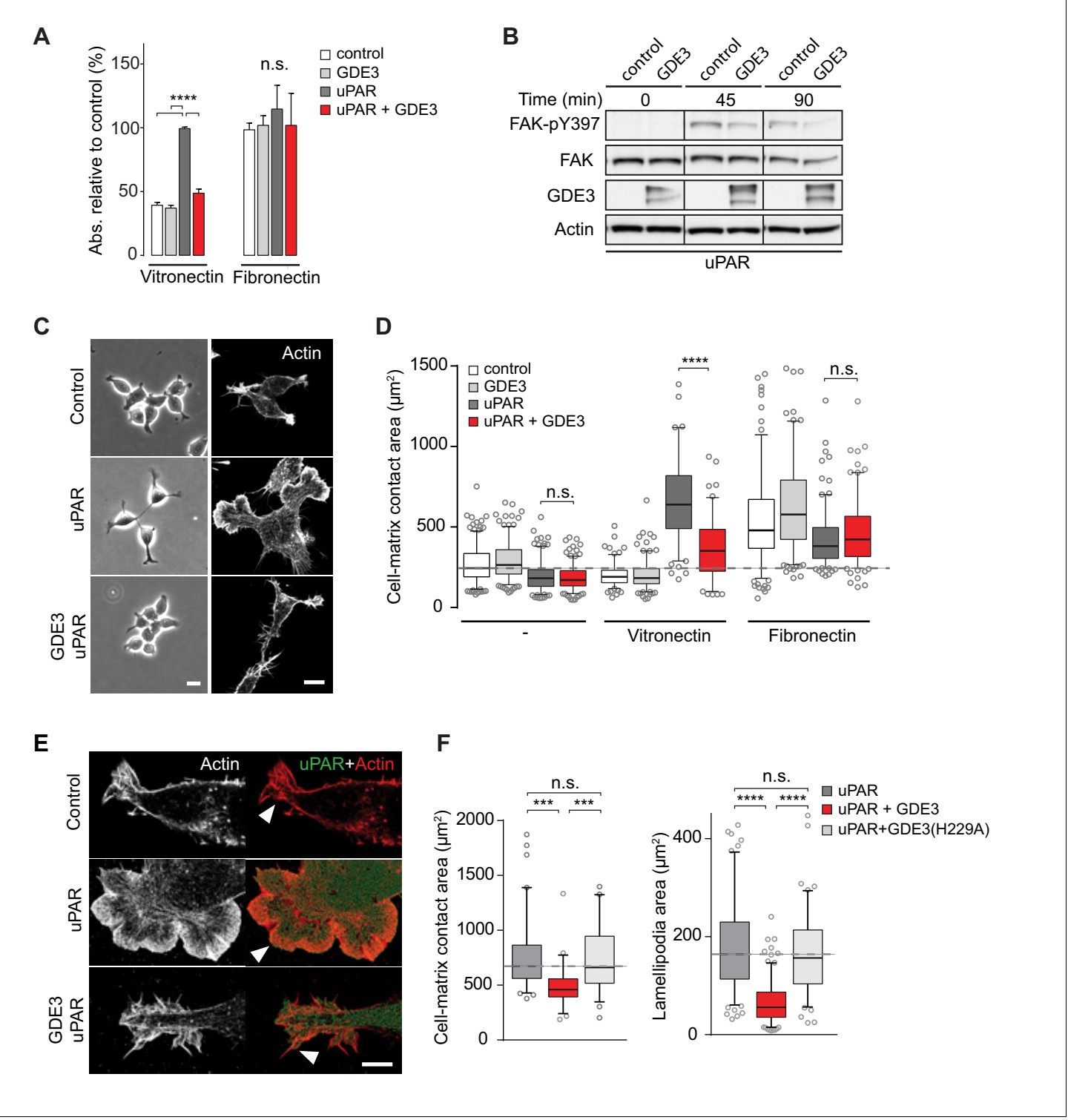

**Figure 3.** GDE3 suppresses uPAR activity in HEK-uPAR cells. (**A**) uPAR confers increased adhesion to vitronectin, but not fibronectin, which is prevented by GDE3 expression (n = 3, mean ±SEM), ****p<0.0001; n.s., not significant. (**B**) GDE3 inhibits FAK activation during cell adhesion. FAK activity was assayed at the indicated times after plating. (**C**) GDE3 inhibits uPAR-induced cell spreading, scattering and lamellipodia formation on vitronectin (bar, 10 μm). (**D**) Cell-matrix contact area of cells expressing the indicated constructs plated on either fibronectin or vitronectin; (-) denotes cells on uncoated cover slips. Box plots show the mean of three independent experiments. Dotted line represents the mean of control cells (n = 3, mean ±SEM, ****p<0.0001; n.s., not significant). (**E**) uPAR-induced lamellipodia on vitronectin disappear upon GDE3 expression, as shown by super-resolution

*Figure 3 continued on next page*

*Figure 3 continued*

microscopy (bar, 5 μm). (F) Quantification of cell-matrix contact area in GDE3- and GDE3(H229A)-expressing cells. GDE3(H229A) fails to affect lamellipodia formation (n = 3, mean ±SEM ***p<0.001; ****p<0.0001).

DOI: https://doi.org/10.7554/eLife.23649.006

## GDE3 overexpression attenuates the transformed phenotype, slows tumor growth in vivo and correlates with improved survival in breast cancer

In long-term assays, wild-type MDA-MB-231 cells showed marked scattering, indicative of increased cell motility with loss of intercellular contacts (*Figure 6A*). Again, GDE3 overexpression mimicked uPAR depletion either in greatly reducing both cell motility and clonogenic potential, using either shRNA-mediated knockdown (*Figure 6A,B*) or CRISPR-mediated knockout of uPAR (*Figure 6—figure supplement 1*).

Having shown that GDE3 suppresses the non-proteolytic activities of uPAR, we next examined how GDE3 affects uPAR-driven proteolytic matrix degradation by MDA-MB-231 cells. Also in this cell system, GDE3 overexpression mimicked uPAR silencing in inhibiting the degradation of a gelatin matrix (mixed with vitronectin) in the presence of serum (*Figure 6C,D*) (*Figure 6—figure supplement 1*). On the basis of these results, we conclude that GDE3 attenuates the transformed phenotype of uPAR-positive breast cancer cells through loss of functional uPAR.

When injected into the mammary fat pads of female nude mice, GDE3-overexpressing MDA-MB-231 cells showed diminished tumor growth over time, when compared to empty vector-expressing cells (*Figure 7A*), consistent with the cell-based data. However, the full implications of GDE3 expression on uPAR-dependent tumor growth remain to explored in further detail. Finally, in patients, high expression of *GDPD2* was found to correlate with prolonged relapse-free survival in breast cancer, particularly in triple-negative (basal-like) subtype patients (N = 618) (*Figure 7B*). No such correlation was found for GDE2 (encoded by *GDPD5*; not shown). This suggests that GDE3/GDPD5 may serve as a marker of clinical outcome in breast cancer.

## Discussion

GPI-anchoring is a complex post-translational modification that anchors select proteins in the outer leaflet of the plasma membrane. Despite decades of research, the biological significance of GPI anchors has long remained a mystery (*Kinoshita and Fujita, 2016*; *Paulick and Bertozzi, 2008*). Some GPI-anchored proteins are released from their anchor and detected in body fluids, implying involvement of one or more endogenous GPI-specific hydrolases. Recent studies have advanced the field by showing that cleavage and shedding of certain GPI-anchored proteins is mediated by a cell-intrinsic transmembrane glycerophosphodiesterase, termed GDE2 (or GDPD5), thereby promoting neuronal differentiation through multiple signaling pathways (*Matas-Rico et al., 2016*; *Matas-Rico et al., 2017*; *Park et al., 2013*).

In this study, we focused on the shedding of GPI-anchored uPAR because of its regulatory role in multiple cellular and (patho)physiological activities, while soluble uPAR is considered a biomarker of various human pathologies. Here we report that GDE3 functions as a long-sought GPI-specific PLC that releases uPAR from its anchor. By contrast, its homologue GDE2 failed to release uPAR. As a consequence of GDE3 action, uPAR loses its vitronectin-dependent and matrix-degrading activities, when assayed in HEK293-uPAR and triple-negative breast cancer cells that express both uPAR and uPA. Importantly, loss of uPAR expression by GDE3 was found to be restricted to certain microdomains at the basolateral plasma membrane, where signal transduction is likely to take place. Thus, by acting as a GPI-specific PLC towards uPAR, GDE3 is a negative regulator of the uPAR signaling network (*Figure 7C*) that includes uPAR's proteolytic and non-proteolytic activities. Consistent with this, GDE3 overexpression in uPA/uPAR-positive MDA-MB-231 breast cancer cells slowed tumor progression in a xenograft mouse model. Although statistically significant, the inhibitory effect of GDE3 overexpression on tumor growth was not dramatic, which should not come as a surprise since MDA-MBA-31 cells express the strongly oncogenic mutant K-RAS protein, which tends to override the regulation of numerous signaling pathways. Yet, this finding adds to the relevance of GPI-

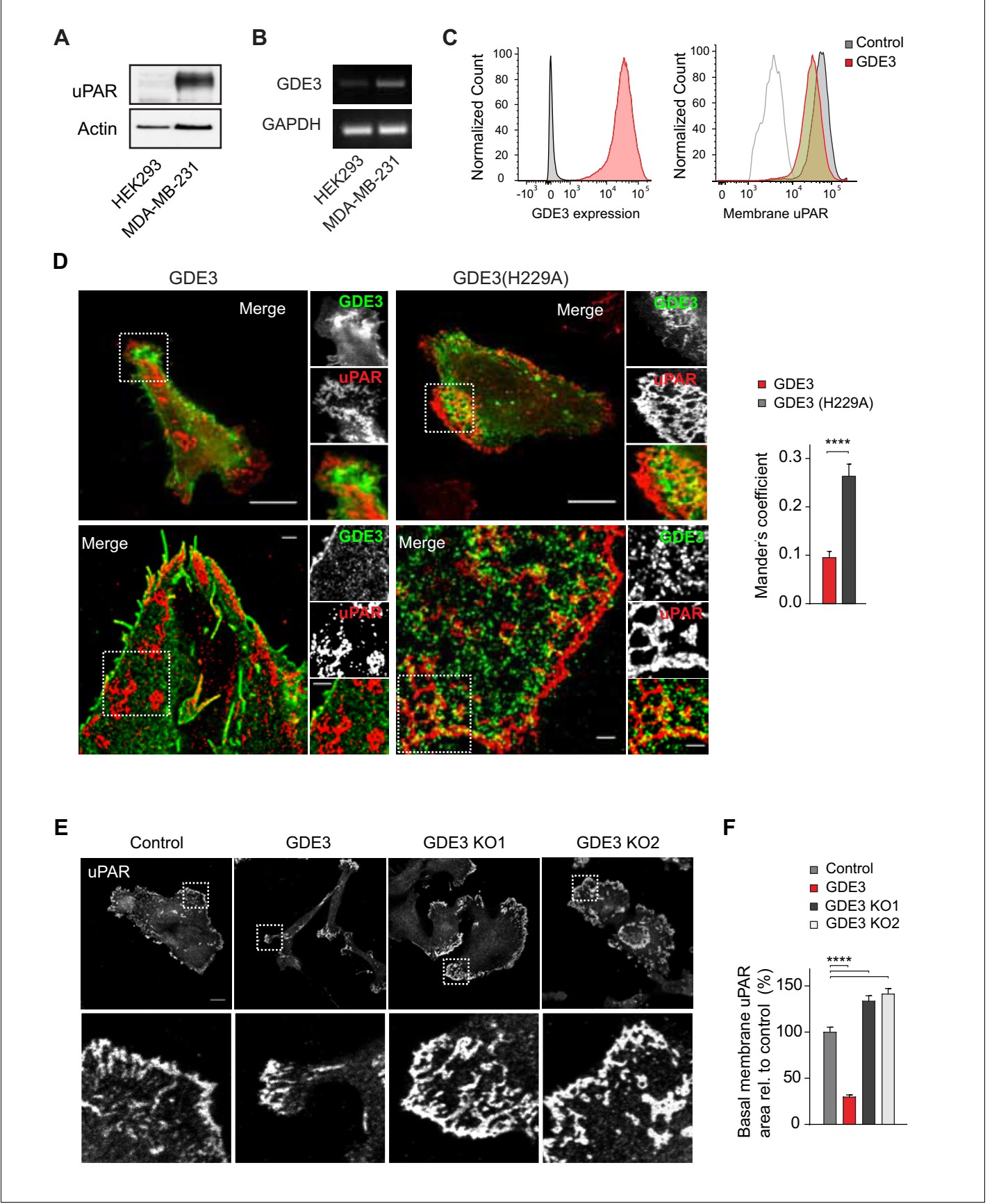

**Figure 4.** GDE3 depletes uPAR from distinct basolateral membrane domains in MDA-MB-231 breast cancer cells. (A) Endogenous uPAR expression in MDA-MB-231 versus HEK293 cells, as determined by immunoblot. (B) Endogenous *GDPD2* expression, as determined by qPCR analysis. (C) (left) Cell-surface expression of GDE3-mCherry of MDA-MB-231 cells expressing GDE3, as detected by flow cytometry. (Right) Cell-surface expression of uPAR in control (grey) and GDE3-expressing MDA-MB-231 cells (red), as detected by flow cytometry. (D) Confocal (top) and dual-color super-resolution microscopy images (bottom) of MDA-MB-231 cells expressing GDE3-GFP or catalytically dead GDE3(H229A)-GFP. Endogenous uPAR was immunostained in red. Merged images show colocalization of uPAR with GDE3(H229A) but not with wild-type GDE3 and uPAR. Scale bars, 10 μm (confocal) and 1 μm (super-resolution). Co-localization analysis (Mander's coefficient) on peripheral uPAR patches in confocal images was done using ImageJ software (n = 30 cells, three independent experiments). (E) Endogenous uPAR staining in control, GDE3-overexpressing and GDE3 knockout MDA-MB-231 cells plated on vitronectin. Two distinct GDE3 knockout clones (KO1 and KO2) were examined, as indicated. Scale bar,10 μm. (F) Quantification of basolateral uPAR-containing membrane domains referring to the cells in panel (E) (n = 3, mean ±SEM, ****p<0.0001). GDE3 suppresses the vitronectin- and uPAR-dependent phenotype of MDA-MB-231 breast cancer cells.

DOI: https://doi.org/10.7554/eLife.23649.007

The following figure supplement is available for figure 4:

**Figure supplement 1.** GDE3 knockout validation.

DOI: https://doi.org/10.7554/eLife.23649.008

---

specific phospholipases in slowing tumor progression. Furthermore, high GDE3 expression was found to correlate with increased survival probability in triple-negative breast cancer patients. Interestingly, our previous work revealed a similar association between overexpression of GDE2 and positive clinical outcome in neuroblastoma patients, which appears attributable to GDE2-induced glypican shedding(*Matas-Rico et al., 2016*). The present patient survival analysis should be interpreted with caution, however, since involvement of uPAR release remains to be formally proven. Furthermore, we cannot rule out that GDE3 may cleave additional GPI-anchored substrates whose functional loss could contribute to positive clinical outcome.

The present results predict that, depending on its expression levels, GDE3 may downregulate normal uPAR-dependent remodeling processes. Indeed, upregulated GDE3 accelerates osteoblast differentiation (*Corda et al., 2009*; *Yanaka et al., 2003*) in a manner resembling the uPAR knockout phenotype (*Furlan et al., 2007*). Furthermore, a striking >200 fold upregulation of *GDE3/GDPD2* is observed during blastocyst formation (*Munch et al., 2016*), implicating GDE3 in the invasion of pre-implantation embryos, a process in which the uPA/uPAR signaling network has been implicated (*Multhaupt et al., 1994*; *Pierleoni et al., 1998*). Although correlative, these results support the view that GDE3 is upregulated to downregulate uPAR activity in vivo. The present findings also suggest that circulating full-length suPAR should be regarded as a marker of GDE3 activity, not necessarily reflecting uPAR expression levels.

It will now be important to determine how GDE3 expression and activity are regulated and, furthermore, to explore the substrate selectivity of the respective GDEs in further detail. Homology modeling revealed striking differences in electrostatic surface properties of GDE2 versus GDE3, suggesting that protein-protein interactions may determine substrate recognition by these GDE family members. Specific GPI-anchor modifications (*Kinoshita and Fujita, 2016*; *Paulick and Bertozzi, 2008*) could also determine the sensitivity of GPI-anchored proteins to GDE attack.

Finally, when regarded in a broader context, the present and previous findings (*Matas-Rico et al., 2016*; *Matas-Rico et al., 2017*; *Park et al., 2013*) support the view that vertebrate GDEs, notably GDE2 and GDE3, have evolved to modulate key signaling pathways and alter cell behavior through selective GPI-anchor cleavage.

## Materials and methods

### Cell culture and materials

HEK293, MDA-MB-231 and Hs578T cells were obtained from the ATCC and grown in Dulbecco's modified Eagle's medium (DMEM) supplemented with 10% fetal bovine serum (FBS) and antibiotics at 37°C under 5% CO2. Original MDA-MB-231 cells were pathogen tested using the Impactl test (Idexx Bioresearch, Westbrook, ME, USA) and were negative for all pathogens tested. All cell lines were routinely tested negative for mycoplasma contamination. Antibodies used: anti-mCh and anti-GFP, home-made; anti-Flag, M2, anti-Vinculin and β-Actin (AC-15) from Sigma; anti-uPAR (MAB807) from R&D systems; anti-uPAR (13F6) (*Zhao et al., 2015*); anti-FAK(pTyr397) from Thermo Fisher.

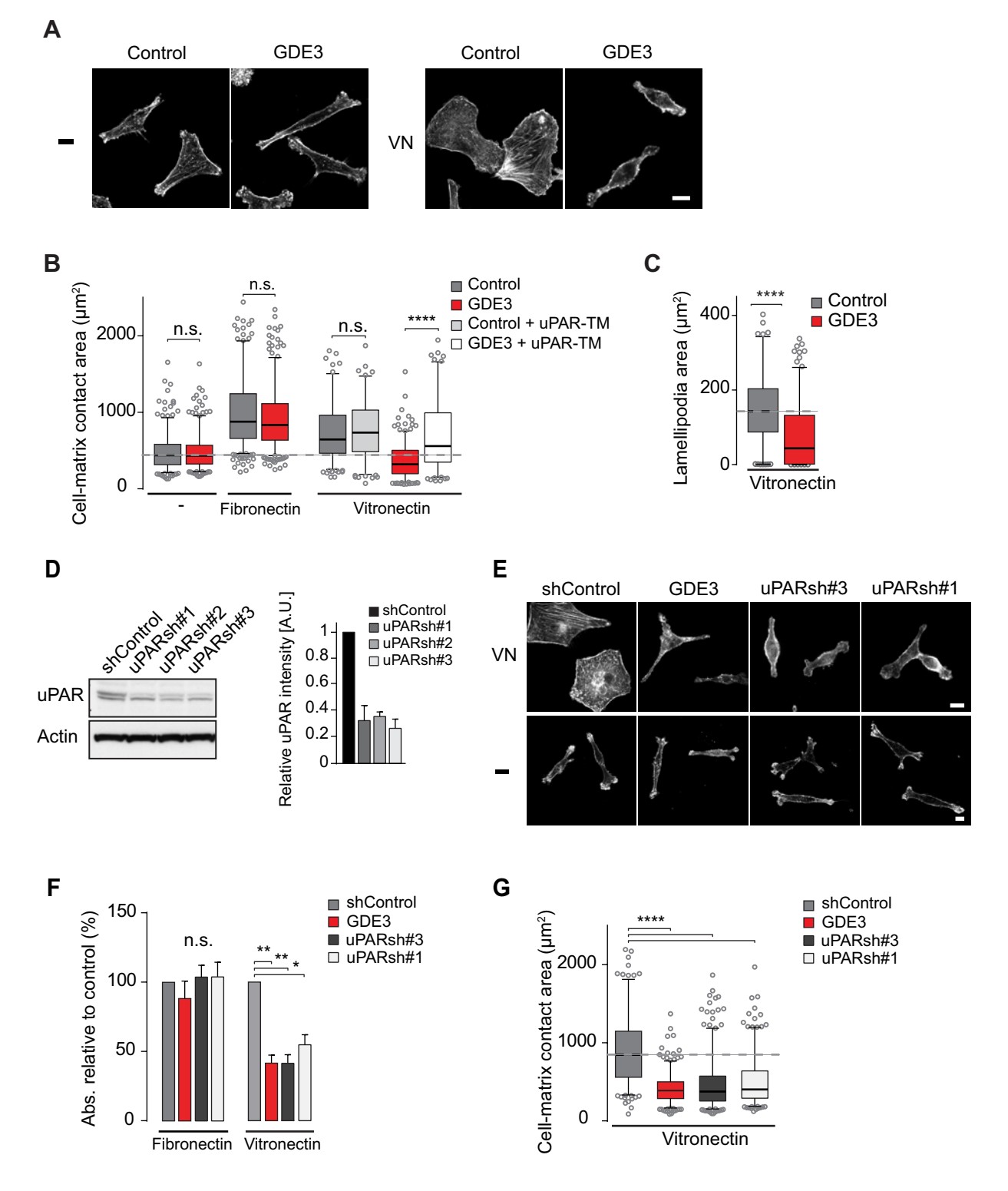

**Figure 5.** GDE3 suppressess the vitronectin- and uPAR-dependent transformed phenotype of MDA-MB-231 breast cancer cells. A) Confocal images showing that GDE3 prevents cell spreading and lamellipodia formation on vitronectin (VN) but not on uncoated cover slips (-). Bar, 10 μm. (B) Quantification of reduced cell spreading on vitronectin by GDE3. Non-cleavable uPAR-TM prevents GDE3 attack. ****p<0.0001; n.s., not significant. (C) Quantification of lamellipodia formation on vitronectin. ****p<0.0001. (D) Immunoblot analysis of shRNA-mediated uPAR knockdown; maximum knockdown was achieved by small hairpins #1 and #3. The upper protein band represents full-length uPAR, the lower band proteolytically cleaved uPAR

*Figure 5 continued on next page*

*Figure 5 continued*

(D2 +D3) (*Høyer-Hansen et al., 1992*). (**E**) GDE3 overexpression mimics the uPAR knockdown phenotype in cells plated on vitronectin (VN); bar, 10 µm. (-) denotes cells on non-coated cover slips. (**F,G**) Quantification of cell adhesion (**F**) n = 3; mean ±SEM) and cell spreading (**G**) induced by GDE3 and uPAR knockdown on the indicated substrates. *p<0.05 **p<0.01; ****p<0.0001. GDE3 overexpression attenuates the uPAR-dependent transformed phenotype of breast cancer cells.

DOI: https://doi.org/10.7554/eLife.23649.009

The following figure supplement is available for figure 5:

**Figure supplement 1.** GDE3 overexpression suppresses the uPAR-vitronectin-dependent phenotype in Hs578T breast cancer cells.

DOI: https://doi.org/10.7554/eLife.23649.010

Vitronectin, fibronectin, inositol 1-phosphate (dipotassium salt) and inositol were purchased from Sigma-Aldrich. *B. cereus* PI-PLC was from Molecular Probes. Phalloidin red (actin-stain 647

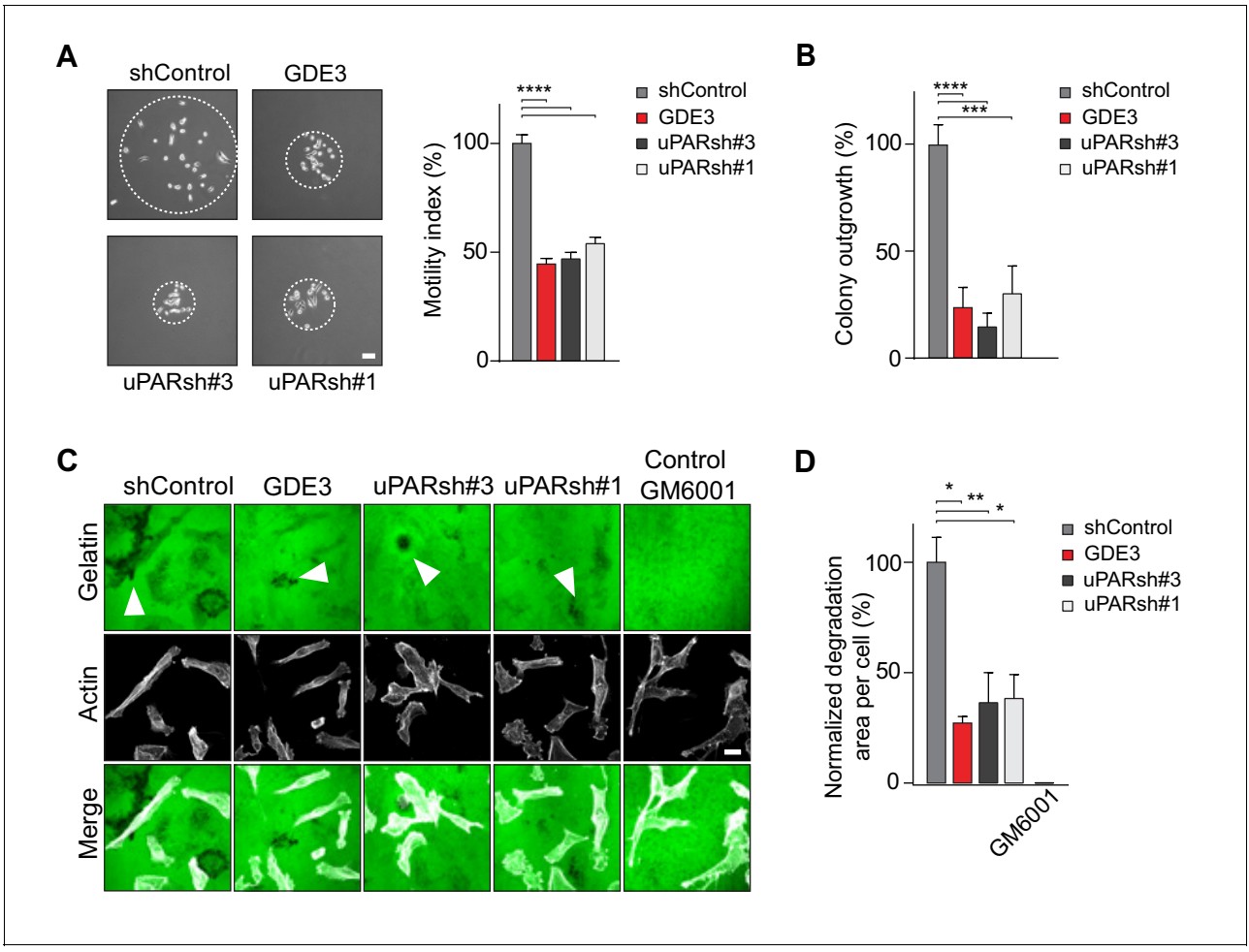

**Figure 6.** GDE3 attenuates the uPAR-dependent transformed phenotype of breast cancer cells. (**A**) GDE3 overexpression in MDA-MB-231 cells mimics uPAR knockdown in inhibiting cell scattering. MDA-MB-231 cells grow out as scattered colonies; bar, 100 µm (n = 3, mean ±SEM; ***p<0.001; ****p<0.0001). (**B**) GDE3 overexpression mimics uPAR knockdown in suppressing colony formation. colonies were counted after 14 days (n = 3, mean ±SEM; ***p<0.001; ****p<0.0001. (**C**) Degradation of a gelatin matrix mixed with vitronectin by MDA-MB-231 cells in the presence of serum (bar, 10 µm). Arrows point to black spots where gelatin was degraded. Metalloprotease inhibitor GM6001 was used as a control. (**D**) Quantification of matrix degradation at 20 hr after plating (mean ±SEM; *p<0.05; **p<0.01).

DOI: https://doi.org/10.7554/eLife.23649.011

The following figure supplement is available for figure 6:

**Figure supplement 1.** uPAR knockout in MDA-MB-231 cells phenocopies GDE3 overexpression.

DOI: https://doi.org/10.7554/eLife.23649.012

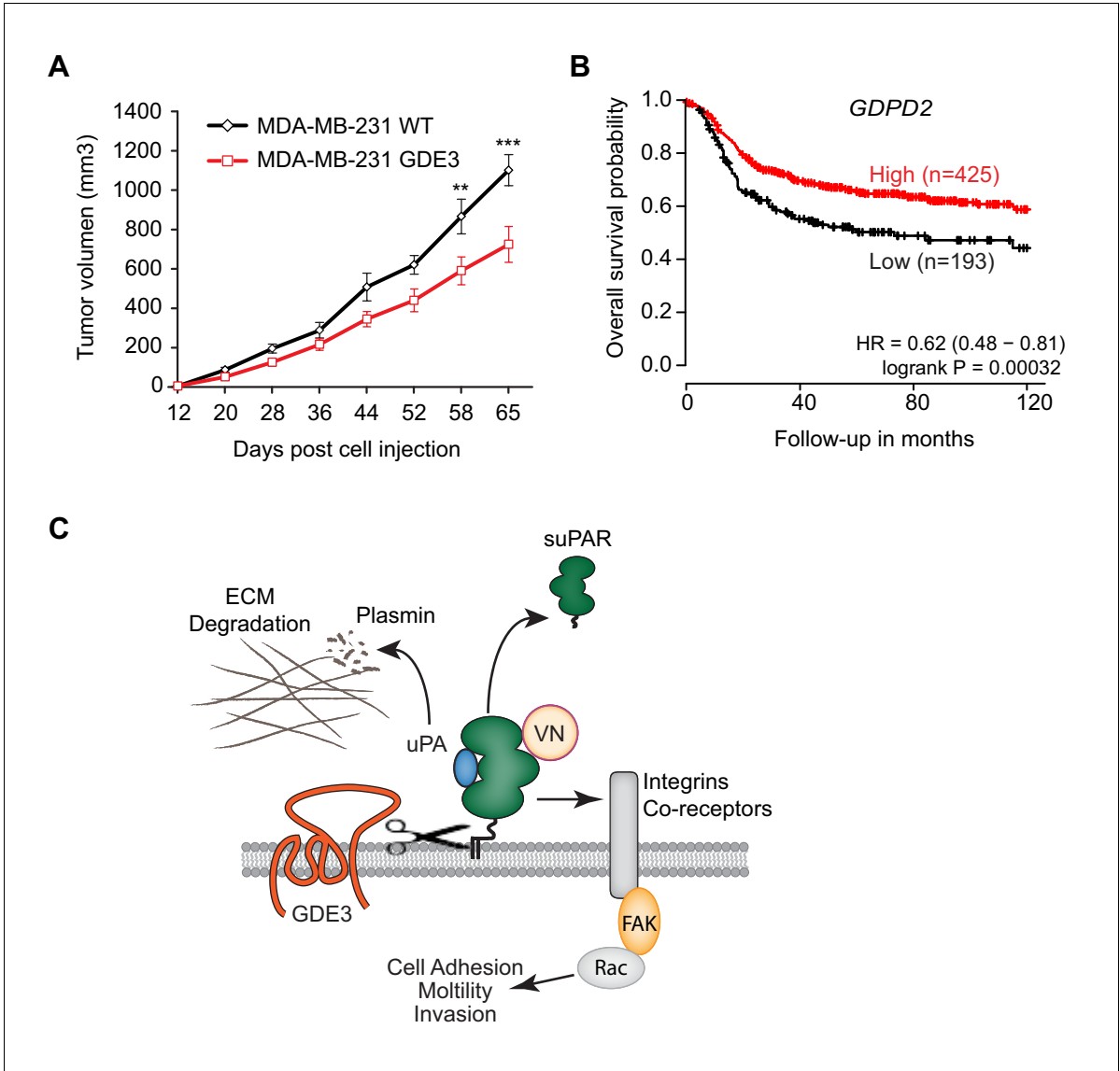

**Figure 7.** GDE3 overexpression in breast cancer cells slows tumor growth in mice, correlating with higher survival probability in patients. (A) Female nude mice (n = 16) were injected with either wild-type or GDE3-overexpressing MDA-MB-231 cells into the mammary fat pad, as described under Materials and methods. Tumor volume was measured every three days after injection for 9 weeks until the tumor had grown to the appropriated size (data represent the mean ±SEM; **p<0.01; ***p<0.001, t-test corrected for multiple comparison). (B) High GDPD2 expression significantly correlates with higher survival rate in triple-negative (basal-like) subtype breast cancer patients (N = 618). HR, hazard ratio with 95% confidence interval indicated. Analysis based on microarray data (www.kmplot.com). (C) Schematic model. GDE3 functions as a GPI-PLC that cleaves and sheds uPAR, leading to loss of uPAR function. VN, vitronectin, FAK, focal adhesion kinase, suPAR, soluble uPAR.

DOI: https://doi.org/10.7554/eLife.23649.013

phalloidin) and green (actin-stain 488 phalloidin) were from Cytoskeleton. GM6001 was from Millipore. Research Source Identifiers: MDA-MB-231 cells RRID:CVCL_0062; Hs578T cells RRID:CVCL_0332; Antibodies: Flag M2 RRID:AB_259529; Anti Vinculin RRID:AB_10746313; Anti actin RRID:AB_2223210; uPAR RRID:AB_2165463.

## Expression vectors

GDE2 cDNA was subcloned as described (*Matas-Rico et al., 2016*). GDE3 cDNA was amplified by PCR and subcloned into a pcDNA3-HA plasmid using AflII/HpaI (PCR product) and AflII/EcoRV (plasmid) restriction sites. GDE3 was recloned into a pcDNA3(-mCherry) construct by PCR amplification

with restriction sites Pml1/Xba1, followed by vector digestion using EcoRV/Xba1. GDE6 transcript variant X7 (GDPD4; NCBI: XM_011544834.1) was cloned into pcDNA5_FRT_TO_puro (provided by dr. Geert Kops, University Medical Center Utrecht). All constructs were epitope- tagged on the C-terminus unless otherwise stated. Mutant GDE3(H229A) was generated by amplification with oligos containing the mutation, followed by Dpn1 digestion of the template. The viral plasmids (pBABE-GDE3-mCh, pBABE-uPAR-GFP) were constructed by subcloning the GDE3-pcDNA3 and uPAR-GFP-pEGFP-N1 into a pBABE plasmid. GDE3-mCherry-pcDNA3 was cut using PmeI followed by digestion of the pBABE backbone with SnaBI. uPAR-GFP-pEGFP-N1 was cut with BglII and HpaI and ligated into the pBABE vector, digested with BAMHI and SnaB1. Constructs uPAR-GFP, uPAR-FLAG and non-cleavable uPAR-TM were previously described (*Caiolfa et al., 2007*; *Cunningham et al., 2003*). Transmembrane-anchored uPAR-TM was constructed by substituting the GPI-anchoring sequence of uPAR (aa 274–313) with the transmembrane region (aa 614–653) of the human epidermal growth factor receptor (EGFR), as described (*Cunningham et al., 2003*).

## Transfections and RNA$_i$-mediated knockdown

Cells stably expressing uPAR-GFP or GDE3-mCherry were generated using retroviral transduction and subsequent selection with puromycin. Transient transfections were done using the calcium phosphate protocol or XtremeGene 9 agent (Roche). Stable uPAR knockdown in MDA-MB-231 cells was achieved using shRNAs in a lentiviral pLKO vector; five shRNAs from three RC human shRNA library were tested: TRCN0000052637, TRCN0000052636, TRCN0000052634, TRCN0000052633 and TRCN0000052635. The latter two were used for experiments; sequences: CCGGCCCATGAATCAA TGTC TGGTACTCGAGTACCAGACATTGATTCATGGGTTTTTG and CGGGCTTGAAGA TCAC-CAGCCTTACTCGAGTAAGGCTGGTGATCTTCAAGCTTTTTG, respectively. For virus production, HEK293T cells were transiently transfected using calcium phosphate, and virus particles were collected 48 hr thereafter. uPAR knockdown cells were selected in medium containing 2 µg/ml puromycin.

## CRISPR knockout

uPAR and GDE3 knockout cell lines were generated using CRISPR/Cas9 genome editing. CRISPR sequences were designed targeting uPAR (*PLAUR; exon 2; 5'- CATGCAGTGTAAGACCAACG-3' and 5'-CCAGGGCGCACTCTTCCACA-3'*) or GDE3 (*GDPD2; exon 2; 5'-AGGATGCAAACCAG-CAAGG-3'*) and cloned into pX330 (*Cong et al., 2013*). MDA-MB-231 cells were transfected with the exon-specific pX330 plasmids in addition to a plasmid containing a guide RNA to the *Danio Rerio* TIA gene (5'- GGTATGTCGGGAACCTCTCC -3') and a cassette of a 2A sequence followed by a BlastR gene, flanked by two TIA target sites. Co-transfection results in infrequent integration of the BlastR gene at the targeted genomic locus by NHEJ, as previously described (*Blomen et al., 2015*). Successful integration of the cassette renders cells resistant to blasticidin. Three days following transfection, the culture medium was supplemented with blasticidin (25 µg/ml). Surviving colonies were clonally expanded, screened for cassette integration and indels into the query gene by PCR (GDE3; 5'- TATGAATCCTGCCCGAAAAG-3' and 5'-AGAGCAGGCCAAACCAGATA-3') or by western blot analysis of the target protein (uPAR).

## Liquid chromatography-mass spectrometry (LC-MS)

To determine the inositol phosphate content of cleaved uPAR, suPAR was immuno-precipitated from HEK293 cell conditioned medium using anti-GFP beads (ChromoTek). To remove inositol phosphate from suPAR, the beads were treated with 0.1M acetate buffer (pH 3.5) and subsequently with 0.5M NaNO2 or 0.5M NaCl (Control) for 3 hr as previously described (*Mehlert and Ferguson, 2009*). Inositol phosphate-containing samples were preprocessed by adding methanol to a final concentration of 70% and shaken at 1000 RPM at room temperature for 10 min. Following centrifugation (20,400 x *g* at 4°C for 10 min), the supernatant was evaporated to dryness in a Speedvac at room temperature. The dried extracts were reconstituted in 50 mM ammonium acetate (pH 8.0), centrifuged (20,400 x *g* at 4°C for 10 min) and transferred to autosampler vials. Liquid chromatography (LC) was performed using a Dionex Ultimate 3000 RSLCnano system (Thermo Fisher Scientific). A volume of 5 µl was injected on a Zorbax HILIC PLUS column (150 × 0.5 mm, 3.0 µm particles) maintained at 30°C. Elution was performed using a gradient: (0–5 min, 20% B; 5–45 min 20–100% B;

45–50 min 100% B; 50–50.1 min 20% B; 50.1–60 min 20% B) of 100% acetonitrile (mobile phase A) and 50 mM ammonium acetate adjusted to pH 8.0 with ammonium hydroxide (mobile phase B) at a flow rate of 15 µl/min. Inositol 1-phosphate was detected with an LTQ-Orbitrap Discovery mass spectrometer (Thermo Fisher Scientific) operated in negative ionization mode scanning from $m/z$ 258 to 260 with a resolution of 30,000 FWHM. Electrospray ionization was performed with a capillary temperature set at 300°C and the sheath, auxiliary and sweep gas flow set at 17, 13 and 1 arbitrary units (AU), respectively. Setting for Ion guiding optics were: Source voltage: 2.4 kV, capillary voltage: −18 V, Tube Lens: −83 V, Skimmer Offset: 0 V, Multipole 00 Offset: 5 V, Lens 0: 5 V, Multipole 0 Offset: 5.5 V, Lens 1: 11 V, Gate Lens Offset: 68 V, Multipole 1 Offset: 11.5 V, Front Lens: 5.5 V. Data acquisition was performed using Xcalibur software (Thermo Fisher Scientific). Reference inositol phosphate (Sigma) was used to determine the retention on the Zorbax HILIC plus column. After applying Xcalibur's build-in smoothing algorithm (Boxcar, 7), extracted ion chromatograms ($m/z$ 259.02–259.03) were used to semi-quantitatively determine inositol phosphate levels.

## Cell adhesion and spreading

48-wells plates were coated overnight at 4°C with fibronectin (10 µg/ml) or vitronectin (5 µg/ml), or left untreated. Thereafter, plates were blocked for 2 hr at 37°C using 0.5% BSA in PBS. Cells were washed and harvested in serum-free DMEM supplemented with 0.1% BSA. Equal numbers of cells were seeded and allowed to adhere for 1 hr. Non-adherent cells were washed away using PBS. Attached cells were fixed with 4% paraformaldehyde (PFA) for 10 min, followed by washing and staining with Crystal violet (5 mg/ml in 2% ethanol) for 10 min. After extensive washing, cells were dried and lysed in 2% SDS for 30 min. Quantification was done by measuring absorbance at 570 nm using a plate reader. For cell-matrix contact area and lamellipodia measurements, coverslips were coated overnight with fibronectin or vitronectin and washed twice with PBS. Cells were trypsinized, washed and resuspended in DMEM and left to adhere and spread for 4 hr. After fixation (4% PFA) and F-actin staining with phalloidin, images were taken using confocal microscopy. Cell and lamellipodia area was quantified using an ImageJ macro.

## Cell scattering and colony formation

Cell scattering was determined as described (*LeBeau et al., 2013*). In brief, single MDA-MB-231 cells were allowed to grow out as colonies, and the area covered by the scattered colonies (colony size) was measured at 6 days after plating. For measuring colony outgrowth, 500 cells were plated and colonies were counted after 14 days.

## Matrix degradation

Coverslips were coated with OG-labelled gelatin (InVitrogen) supplemented with vitronectin (5 µg/ml). About 100.000 cells per coverslip were seeded in DMEM with 10% FCS. After 20 hr, cells were washed, fixed with 4% PFM and stained with phalloidin-Alexa647 (InVitrogen). Gelatin degradation was determined from confocal images of >15 randomly automatically chosen fields of view per coverslip (testing at least two coverslips/condition on two separate days: total four coverslips per condition). The images were randomized and the area of degradation was normalized to the total area of cells or to the number of cells.

## RT-qPCRT

Total RNA was isolated using the GeneJET purification kit (Fermentas). cDNA was synthesized by reverse transcription from 2 µg RNA with oligodT 15 primers and SSII RT enzyme (Invitrogen). Relative qPCR was measured on a 7500 Fast System (Applied Biosystems) as follows: 95°C for 2 min followed by 40 cycles at 95°C for 15 s followed by 60°C for 1 min. 200 nM forward and reverse primers, 16 µl SYBR Green Supermix (Applied Biosystems) and diluted cDNA were used in the final reaction mixture. GAPDH was used as reference gene and milliQ was used as negative control. Normalized expression was calculated following the equation NE = 2(Ct target-Ct reference). Primers used: GDE3, forward TCAGCAGGACCACGAATGTA, reverse GCTGCAGCTTCCTCCAATAG; uPAR, forward AATGGCCGCCAGTGTTACAG, reverse CAGGAGACATCAATGTGGTTC; Cyclophilin, forward CATCTGCACTGCCAAGACTGA, reverse TTGCCAAACACCACATGCTT. For RT-PCR 25 ng cDNA was used in a RT-PCR reaction using GoTaq (Promega). Primer sequences GAPDH forward 5'-CCA

TGTTCGTCATGGGTGT-3', GAPDH reverse 5'-CCAGGGGTGCTAAGCAGTT-3', GDE3 forward 1 5'-TGTTTGAGACTGATGTGATGGTC-3', GDE3 reverse 1 5'-TTCGGGTTGGGAATACAGAG-3'

## Western blotting

For Western blotting, cells were washed with cold PBS, lysed in RIPA buffer supplemented with protease inhibitors and spun down. Protein concentration was measured using a BCA protein assay kit (Pierce) and LDS sample buffer (NuPAGE, Invitrogen) was added to the lysate or directly to the medium. Equal amounts were loaded on SDS-PAGE pre-cast gradient gels (4–12% Nu-Page Bis-Tris, Invitrogen) followed by transfer to nitrocellulose membrane. Non-specific protein binding was blocked by 5% skimmed milk in TBST; primary antibodies were incubated overnight at 4°C in TBST with 2% skimmed milk. Secondary antibodies conjugated to horseradish peroxidase (DAKO, Glostrup, Denmark) were incubated for 1 hr at room temperature; proteins were detected using ECL Western blot reagent.

## Triton X-114 phase separation

HEK293 cells transiently transfected with GDE2, GDE3 or empty plasmid, were plated on PEI-coated 6-well plates. After 24 hr complete medium was replaced with 1 ml serum free medium, 24 hr thereafter the conditioned medium was collected. 2% pre-condensed Triton X-114 was added to ice-cold conditioned medium and phases were separated as previously described (*Doering et al., 2001*). The top aqueous phase and the bottom detergent phase were analyzed by SDS-PAGE and Western blotting.

## Microscopy

Cells cultured on 24 mm, #1,5 coverslips were washed and fixed with 4% PFA, permeabilized with 0.1% Triton X-100 and blocked with 2% BSA for 1 hr. Incubation with primary antibodies was done for 1 hr followed by incubation with Alexa-conjugated antibodies or Phalloidin for 45 min at room temperature. For confocal microscopy, cells were washed with PBS, mounted with Immnuno-MountTM (Thermo Scientific) and visualized on a LEICA TCS-SP5 confocal microscopy (63 x objective). Super-resolution imaging was done using a SR-GSD Leica microscope equipped with an oxygen scavenging system, as previously described (*Matas-Rico et al., 2016*). In short, 15000 frames were taken in TIRF mode, at 10 ms exposure time. After post image analysis, movies were analyzed and corrected using the ImageJ plugin Thunderstorm (http://imagej.nih.gov/ij/) followed by correction with an ImageJ macro using the plugin Image Stabilizer. For Total Internal Reflection (TIRF) microscopy, HEK293 cells stably expressing UPAR-GFP and transiently transfected with GDE3-mCherry were imaged using a Leica AM TIRF MC microscope with a HCX PL APO 63x, 1.47 NA oil immersion lens. Excitation was at 488 and 561 nm and detection of fluorescence emission was by a GR filter cube (Leica). Before each experiment, automatic laser alignment was carried out and TIRF penetration depth was set to 200 nm. Data were acquired at 500 ms frame rate. Basolateral uPAR patches were visualized using confocal microscopy and the area was quantified using an ImageJ macro that measured the uPAR patches based on fluorescence intensity.

## Flow cytometry

HEK293 cells stably expressing uPAR-GFP were left untreated, treated with PI-PLC or transiently transfected with GDE3-mCherry. MDA-MB-231 cells were stably transfected with GDE3-mCherry. Cells were trypsinized, blocked in 2%BSA and stained with rabbit anti-GFP primary antibody or mouse anti-uPAR (13F6) antibody followed by AlexaFluor-647 coupled anti-rabbit secondary antibody. Cells were analyzed using a BD LSR Fortessa flow cytometer.

## Mouse xenograft assay

Equal groups (n = 16) of eight-week-old female NMRI nude mice were injected subcutaneously into the mammary fat pads. Prior to injection, GDE3-expressing or control MDA-MB-231 cells (5 × 105) were suspended in an equal volume of cold phosphate buffered saline and diluted 1:1 with Matrigel. The tumor growth was monitored 3 times a week for 9 weeks by caliper measurements. The tumor volumes were calculated with the formula: 0,5 x length x width¬2 and statistical analysis of the tumor growth was done using a multiple t-test corrected for multiple comparison. A P value < 0.05 was

considered statistically different. The mouse experiments were approved by the Animal Ethics Committee of the Netherlands Cancer Institute (protocol number: 30100 2015 407 appendix 1 WP 6061).

### Statistical analysis

For all single comparisons, a two-tailed unpaired Student's t-test was used; for multiple comparisons, an ordinary ANOVA with Tukey's test was used. A $P$ value $< 0.05$ was considered statistically significant. Error bars shown in the bar diagrams were calculated as the standard error of the mean (SEM); whiskers in the box plots depict 95% confidence intervals.

## Acknowledgements

We thank the people from the Preclinical Intervention Unit of the Mouse Clinic for Cancer and Ageing (MCCA) at the NKI for performing the animal experiments. This work was supported by the Netherlands Organisation for Scientific Research (NWO) and the Dutch Cancer Society (KWF).

## Additional information

### Funding

| Funder | Grant reference number | Author |
| --- | --- | --- |
| Nederlandse Organisatie voor Wetenschappelijk Onderzoek | Graduate Student Fellowship | Daniela Leyton-Puig |
| KWF Kankerbestrijding | Graduate Student Fellowship | Katarzyna M Kedziora |

The funders had no role in study design, data collection and interpretation, or the decision to submit the work for publication.

### Author contributions

Michiel van Veen, Elisa Matas-Rico, Conceptualization, Data curation, Formal analysis, Validation, Investigation, Methodology; Koen van de Wetering, Daniela Leyton-Puig, Katarzyna M Kedziora, Formal analysis, Investigation, Methodology; Valentina De Lorenzi, Yvette Stijf-Bultsma, Investigation, Methodology; Bram van den Broek, Methodology; Kees Jalink, Supervision, Investigation, Methodology; Nicolai Sidenius, Resources, Methodology, Writing—review and editing; Anastassis Perrakis, Formal analysis, Supervision, Validation, Investigation, Methodology; Wouter H Moolenaar, Conceptualization, Data curation, Formal analysis, Supervision, Funding acquisition, Investigation, Methodology, Writing—original draft, Writing—review and editing

### Author ORCIDs

Anastassis Perrakis (iD) http://orcid.org/0000-0002-1151-6227
Wouter H Moolenaar (iD) http://orcid.org/0000-0001-7545-198X

### Ethics

Animal experimentation: The mouse experiments were approved by the Animal Ethics Committee of the Netherlands Cancer Institute (protocol number: 30100 2015 407 appendix 1).

### Decision letter and Author response

Decision letter https://doi.org/10.7554/eLife.23649.015
Author response https://doi.org/10.7554/eLife.23649.016

## Additional files

### Supplementary files

• Transparent reporting form
DOI: https://doi.org/10.7554/eLife.23649.014

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
