## [Decision Letter]

Thank you for submitting your article "Negative regulation of uPAR activity by a GPI-specific phospholipase C" for consideration by *eLife*. Your article has been favorably evaluated by Jonathan Cooper (Senior Editor) and three reviewers, one of whom is a member of our Board of Reviewing Editors. The following individuals involved in review of your submission have agreed to reveal their identity: Michael Ploug (Reviewer #2); Reinhard Fässler (Reviewer #3).

The reviewers have discussed the reviews with one another and the Reviewing Editor has drafted this decision to help you prepare a revised submission.

GPI-membrane-bound uPAR can be cleaved from the cell surface to produce a soluble form (soluble uPAR – suPAR), which is readily detected in blood and urine and is markedly increased in pathological conditions, such as inflammation and in various types of cancer. The mechanism of uPAR release and its physiological implications are still largely unknown. The manuscript by van Veen et al., reports the novel finding that the transmembrane glycerophosphodiesterase GDE3 can promote shedding of uPAR in cultured cells by cleavage of uPAR's GPI-anchor via a phospholipase C-like activity. This finding is interesting because of suPAR's role as a prognostic biomarker for poor patient prognosis. Moreover, increased uPAR expression in solid tumors (and the corresponding activated stroma) is being evaluated by PET-imaging for patient stratification (Persson et al. 2015, Theranostics 5, 1303-1306). However, despite some clear strengths of the submitted manuscript, the research appears to have some important shortcomings and limitations, which the authors should address in a revised submission:

Major points:

1) The bulk of the biochemical data on the shedding of uPAR driven by GDE3 expression/activity performed in transfected HEK293 cells (co-transfection of both uPAR & GDE3) or MDA-MB-231 breast carcinoma cell line (transfection of GDE3). Under these conditions it appears that only a fraction of the total uPAR is actually shed from the cell surface. Consequently, it is recommended the GDE3-mediated release of uPAR should be tested in cell lines that endogenously co-express both uPAR and GDE3 in which GDE3 expression is inhibited or silenced by RNA interference or CRISPR/CAS9 strategies.

2) In their Introduction the authors posit that the mechanism(s) of shedding of soluble uPAR remained enigmatic until their discovery of the potential role of GDE3 in cleaving uPAR's GPI linkage. This is an over-simplification since it is that uPAR can be shed by proteolysis. Indeed, the proteolytically released domain I fragment is actually the more powerful biomarker for poor prognosis and correlates better to cancer burden in a GEM model of breast cancer (Thurison et al., 2015 Mol Carcinog 55: 717-731). Consequently, the authors should clarify the relative role of the different mechanisms of uPAR shedding in the text.

3) In Figure 1 the GDE3-mediated release of uPAR to the medium and the reduction in uPAR at the cell surface (Figure 1) is convincingly shown. However, it also seems that this is a relatively ineffective shedding (despite GDE3 being expressed by transfection). Indeed, the flow cytometry analysis appears to indicate that GDE3 promotes only 10% of the uPAR shedding compared to PI-PLC. This should be quantified and commented on. Moreover, how does the subcellular localization of GDE3 (filopodia, microclusters) compare to that of uPAR in cells that do not so heavily overexpress GDE3 or in the invasive front of solid tumors where uPAR is believed to be concentrated?

4) Figure 1 demonstrates that GDE3 promotes uPAR shedding by a phospholipase C-like mechanism. However, it has been noted by others (Ploug et al., 1991 JBC 266: 1926-1933) that GPI-anchors carrying an additional acylation at the 2-hydroxyl position of the inositol ring would be refractory to the action of GDE3 (and experimentally to GPI-PLC). Since a substantial proportion (~50%) of uPAR may carry this acylation event in some cells, the authors should discuss how this might alter uPAR shedding by the GDE3 mechanism in some cells.

5) In Figure 4 the authors investigate the effect of GDE3 transfection on MDA-MB-231 cells ability to cleave gelatin (denatured collagen). This is an unusual approach since it is well established that uPAR functions by focalizing uPA-mediated plasminogen activation to the cell surface and that the physiologically important substrate for this is fibrin – not collagen (Connolly et al., 2010 Blood 116: 1593-1603). Moreover, the control for inhibition of this activity should not be a broad spectrum MMP inhibitor but alpha2-antiplasmin or prevention of uPA binding by adding a neutralizing antibody or an inactivated variant of uPA (DFP-treated uPA, active site mutated (S356A) or simply the uPAR binding fragment ATF.

6) In the second paragraph of the Results and Discussion the authors speculate, based on in silico modeling, that the selectivity of GDE3 for uPAR arises due to an exosite interaction with the protein moiety of uPAR (Figure 1). This appears highly speculative since the authors have not experimentally demonstrated selectivity of GDE3 for uPAR over other other GPI-anchored uPAR homologs in parallel (e.g. CD177, C4.4A, TEX101 or Haldisin). Moreover, the cleavage of the uPAR-GFP-hybrid (Figure 1) and uPAR-FLAG (Figure 1—figure supplement 2) would tend to argue against such specificity. In their Materials and methods they state that these gene fusions were generated as described by Cunningham et al. 2003 (although no GFP version was described in that paper). In the experiments described by Cunningham et al., the FLAG epitope tag or TM-region is fused at position 275 or 277 in uPAR, which eliminates the region of uPAR proximal to the GPI-attachment site (position 283) thereby eliminating the protein sequence that van Veen et al. propose to serve in exosite binding. It is possible that the citation of Cunningham et al. 2003 may be in error and what the authors actually used was a more recent strategy in which GFP is inserted between uPAR and the GPI-anchoring signal (Hellriegel et al. 2011, FASEB J 25, 2882-2897) leaving residues 276-283 of uPAR in close enough proximity to occupy a putative exosite in GDE3. Although potentially interesting, this suggestion is nevertheless also complicated by the fact that this region is completely different between mouse and human uPAR (and should be mirrored by a similar difference in GDE3). If not elaborated on experimentally this speculation should thus be omitted.

7) In the seventh paragraph of the Results and Discussion it states that invasion of pre-implantation embryos is a known uPA/uPAR-dependent process is incorrect. Although uPAR is expressed by giant trophoblasts, the fact that genetic ablation of uPAR has no effect on mouse fertility (or development, or hemostasis) suggests that this system is not essential for the invasion of pre-implantation mouse embryos (Bugge et al. 1995, JBC 270, 16886).

8) It would be interesting, and potentially important, to test the effects of GDE3 over-expression on the in vivo tumorigenicity and/or metastasis of MDA-MB-231 cells in appropriately xenografted mice. More specifically, to what extent does GDE3 over-expression diminish the in vivo tumorigenicity of MDA-MB-231 cells and to what extent can any GDE3-mediated inhibitory effects on tumorigenesis be reversed by the expression of the GDE3 resistant uPAR-TM construct?

[Editors' note: further revisions were requested prior to acceptance, as described below.]

Thank you for resubmitting your work entitled "Negative regulation of urokinase receptor activity by a GPI-specific phospholipase C in breast cancer cells" for further consideration at *eLife*. Your revised article has been favorably evaluated by Jonathan Cooper (Senior editor), a Reviewing editor, and two reviewers.

The revised paper is much improved but the new data have raised additional questions. The paper now provides strong evidence that over-expressing GDE3 (but not GDE2) causes release of a soluble form of uPAR (suPAR). Combined with the in vitro experiments showing GPI-PLC activity of GDE3 but not GDE2 on uPAR, the paper makes a strong case for GDE3 down-regulating uPAR by cleavage of the inositol phosphodiester of the GPI anchor. Consistent with this hypothesis, over-expressed GDE3 opposes uPAR in various functional assays done with HEK-uPAR and MDA-MB-231 cells. The reviewers were concerned, however, that the functional assays could have been done with better controls, such as catalytically-inactive GDE3 and an uncleavable form of uPAR. Experiments to show whether GDE3 is necessary for uPAR cleavage would have been better done in cells with high levels of GDE3 and uPAR gene expression. In addition, the% decrease in uPAR when GDE3 is over-expressed seems rather small, raising questions whether or not uPAR is the primary substrate. However, the original round of review did not bring out these issues prominently, so we are only recommending minor revisions to be clear about the limitations of the results.

1. Results: "Flow cytometry analysis confirmed loss of uPAR from the plasma membrane by GDE3 (Figure 1), while TIRF microscopy revealed that uPAR loss occurred largely at the basolateral plasma membrane, where integrins normally mediate cell-matrix adhesion (Figure 1)."

"Loss" is a bit strong, and basolateral refers to a membrane domain that is larger than that seen by TIRF microscopy. Perhaps revise to:

"Flow cytometry analysis of GDE3 over-expressing cells confirmed decreased levels of uPAR in the plasma membrane (Figure 1), while TIRF microscopy revealed substantial loss of uPAR loss from the ventral surface (Figure 1)."

2. Figure 1: Given the speculative nature of the model, please relabel the dashed yellow line as "proposed substrate interaction border".

3. Figure 4: upper images show confocal images (focal plane unclear, please clarify) and the lower images show super-resolution, which uses TIRF and thus only the basal membrane. The results show that over-expressed WT GDE3, but not catalytically dead GDE3, segregates away from the uPAR signal (lack of co-localization). Please provide quantification of co-localization over multiple cells. In addition, if the conclusions are based on the TIRF and not confocal data, then the conclusion should be modified to make it clear that GDE3 may induce uPAR relocalization rather than (or as well as) cleavage, because only the basal membrane was visualized.

4. Figure 4: Please clarify how the experiments were carried out. In particular, because uPAR may affect ECM composition and vice versa, what ECM were the cells plated on? Please clarify if n=3 is the number of independent experiments. Why wasn't suPAR release measured in these experiments? If suPAR release from MDA cells is as low as for HEK cells in Figure 1, then it would not be feasible to see a decrease in MDA cells deleted for GDE3, but if MDA cells normally release detectable suPAR then the media of the GCE3 knockout cells should be assayed for suPAR.

5. Figure 7: This experiment shows a modest but statistically significant reduction in tumor volume due to GDE3 over-expression. It is unfortunate that catalytically-dead GDE3 or uPAR-TM were not used as controls but we appreciate the time constraints. In light of the limited controls, we agree with the cautious interpretation in the discussion. However, even more caution may be justified because IHC analyses of solid human cancers show that the majority of uPAR is actually not found on the cancer cells themselves (only a few cancers deviate from this) but on the activated tumor microenvironment - mostly comprising immune cells. Therefore, the reviewers recommend adding a further caveat that the full implications of GDE3 expression on uPAR-dependent tumor growth need to be explored in more detail.

6. Please clarify in the methods the exact structure of the uPAR-TM construct. The Sidenius lab made several such constructs with the tag in different positions.

7. Figure 5: The anti-uPAR antibody detects two bands. Are both bands depicting uPAR proteins?

8. Figure 1—figure supplement 1: It is shown that GDE3 competes with exogenous PI-PLC to deplete uPAR. It is unclear if PI-PLC is not working at all if GDE3 is highly expressed or less efficient. An additional lane showing uPAR and GDE3-expressing cells without PI-PLC would clarify this point.

---

## [Author Response]

*Major points:*

*1) The bulk of the biochemical data on the shedding of uPAR driven by GDE3 expression/activity performed in transfected HEK293 cells (co-transfection of both uPAR & GDE3) or MDA-MB-231 breast carcinoma cell line (transfection of GDE3). Under these conditions it appears that only a fraction of the total uPAR is actually shed from the cell surface. Consequently, it is recommended the GDE3-mediated release of uPAR should be tested in cell lines that endogenously co-express both uPAR and GDE3 in which GDE3 expression is inhibited or silenced by RNA interference or CRISPR/CAS9 strategies.*

We have addressed these points as follows:

- We have used CRISPR-based knockout strategies to knockout both GDE3 and uPAR in breast cancer cells, showing that GDE3-deficient MDA-MB-231 cells show the predicted uPAR-dependent phenotype.

- We now show that an independent triple-negative breast cancer cell line (Hs578T, which is uPAR-positive; NCI-60 panel analysis) has the same phenotype as MDA-MB-231 cells upon GDE3 overexpression.

- FACS analysis shows reduced uPAR expression at the MDA-MB-231 cell surface upon GDE3 expression. Loss of uPAR by GDE3 is indeed partial, as shown by dual-color microscopy.

- Confocal and dual-color super-resolution microscopy in TIRF mode show where GDE3 and uPAR localize at the basolateral plasma membrane, and which uPAR fraction is depleted by active GDE3 but not by inactive GDE3(H229A).

*2) In their Introduction the authors posit that the mechanism(s) of shedding of soluble uPAR remained enigmatic until their discovery of the potential role of GDE3 in cleaving uPAR's GPI linkage. This is an over-simplification since it is that uPAR can be shed by proteolysis. Indeed, the proteolytically released domain I fragment is actually the more powerful biomarker for poor prognosis and correlates better to cancer burden in a GEM model of breast cancer (Thurison et al., 2015 Mol Carcinog 55: 717-731). Consequently, the authors should clarify the relative role of the different mechanisms of uPAR shedding in the text.*

Proteolytic fragmentation of uPAR resulting in the release of the domain 1 fragment from cells is in our opinion most correctly referred to as uPAR-cleavage as the majority of the receptor remains anchored to the plasma membrane after the event. For clarity, we now mention proteolytic fragmentation of uPAR in the Introduction, but we believe a further expansion on this point would only add confusion, as we do not investigate proteolytic cleavage of uPAR in this study. Along the same line, we do not present any biomarker data that would allow us to conclude anything on the possible importance of the GDE3-uPAR axis for the power of (s)uPAR and (s)uPAR-fragments as biomarkers. Although very interesting, we believe discussing these issues would be to speculative.

*3) In Figure 1 the GDE3-mediated release of uPAR to the medium and the reduction in uPAR at the cell surface (Figure 1) is convincingly shown. However, it also seems that this is a relatively ineffective shedding (despite GDE3 being expressed by transfection). Indeed, the flow cytometry analysis appears to indicate that GDE3 promotes only 10% of the uPAR shedding compared to PI-PLC. This should be quantified and commented on. Moreover, how does the subcellular localization of GDE3 (filopodia, microclusters) compare to that of uPAR in cells that do not so heavily overexpress GDE3 or in the invasive front of solid tumors where uPAR is believed to be concentrated?*

See point #1 above. New flow cytometry data show that part of endogenous uPAR in MDA-MB-231 breast cancer cells is shed by active GDE3 (Figure 4). In addition, we have used confocal and TRIF dual-color super-resolution microscopy showing that uPAR (both transfected and endogenous) colocalizes with catalytically dead GDE3(H229A) but not with active wild-type uPAR at distinct microdomains at the ventral plasma membrane (Figure 4). Thus, GDE3 activity regulates endogenous uPAR only in select membrane domains, which is consistent with the observation that active GDE3 does not deplete the entire uPAR membrane pool (Figure 4).

*4) Figure 1 demonstrates that GDE3 promotes uPAR shedding by a phospholipase C-like mechanism. However, it has been noted by others (Ploug et al., 1991 JBC 266: 1926-1933) that GPI-anchors carrying an additional acylation at the 2-hydroxyl position of the inositol ring would be refractory to the action of GDE3 (and experimentally to GPI-PLC). Since a substantial proportion (~50%) of uPAR may carry this acylation event in some cells, the authors should discuss how this might alter uPAR shedding by the GDE3 mechanism in some cells.*

This is a somewhat confusing issue. GPI-anchors are modified by inositol acylation during biosynthesis, but it is far from clear if mature GPI-anchored proteins at the plasma membrane still carry this modification. In fact, normal inositol de-acylation could be impaired under overexpression conditions, such in HEK293 cells. GPI-anchor acylation of erythrocyte acetylcholinesterase renders it resistant to bacterial PI-PLC, but not to GPI-PLD (Roberts et al. 1988, JBC 263: 18766-75), but this seems to be an exception. Consistent with previous results (for example, Hoyer-Hansen et al. 2001, Biochem J 358:673), we find that mature uPAR is efficiently shed by PI-PLC, as well as by GDE3, arguing against a GDE3-resistant fraction. Actually, Ploug et al., 1991 (JBC 266:1926) also describe efficient uPAR release by exogenous PI-PLC. That being said, we cannot find evidence for the notion that “a substantial proportion (~50%) of uPAR may carry this acylation”. We therefore prefer not to discuss this hypothesis.

*5) In Figure 4 the authors investigate the effect of GDE3 transfection on MDA-MB-231 cells ability to cleave gelatin (denatured collagen). This is an unusual approach since it is well established that uPAR functions by focalizing uPA-mediated plasminogen activation to the cell surface and that the physiologically important substrate for this is fibrin – not collagen (Connolly et al., 2010 Blood 116: 1593-1603). Moreover, the control for inhibition of this activity should not be a broad spectrum MMP inhibitor but alpha2-antiplasmin or prevention of uPA binding by adding a neutralizing antibody or an inactivated variant of uPA (DFP-treated uPA, active site mutated (S356A) or simply the uPAR binding fragment ATF.*

We agree that collagen is not a direct substrate for plasmin. However, it has long been known that plasmin can degrade collagen indirectly through activation of latent collagenase. See for example, Werb et al., N Engl J Med 296:1017–1023, 1977). In a tumor context, uPA/uPAR knockdown has strong effects on matrigel invasion and tumor cell angiogenesis, See for example: Subramanian et al., Int. J. Oncol. 28:831-6; 2006; Montgomery et al., Breast Cancer Res. 14:R84; 2012. Furthermore, the involvement of the uPAR-uPA signaling axis in Matrigel and gelatin degradation has been documented in various cell systems (Fleetwood, et al., J Immunol, 2014. 192:3540; Kunigal et al., Int J Cancer, 2007. 121:2307; Kargiotis et al., Int J Oncol, 2008. 33: 937; Duriseti et al. J Biol Chem, 2010. 285:26878). Since we focused on regulation of uPAR by GDE3 in breast cancer cells, we reasoned that degradation of gelatin (mixed with vitronectin and serum factors) would be a relevant model system.

To show that GDE3 expression decreases gelatin degradation through uPAR release, we originally used two hairpins against uPAR. We have now confirmed this finding by using CRISPR-based uPAR knockout cells (Figure 6). We used a widely used MMP inhibitor (GM6001) as a control, simply to show that loss of gelatin occurs through degradation.

*6) In the second paragraph of the Results and Discussion the authors speculate, based on in silico modeling, that the selectivity of GDE3 for uPAR arises due to an exosite interaction with the protein moiety of uPAR (Figure 1). This appears highly speculative since the authors have not experimentally demonstrated selectivity of GDE3 for uPAR over other GPI-anchored uPAR homologs in parallel (e.g. CD177, C4.4A, TEX101 or Haldisin). Moreover, the cleavage of the uPAR-GFP-hybrid (Figure 1) and uPAR-FLAG (Figure 1—figure supplement 2) would tend to argue against such specificity. In their Materials and methods they state that these gene fusions were generated as described by Cunningham et al. 2003 (although no GFP version was described in that paper). In the experiments described by Cunningham et al., the FLAG epitope tag or TM-region is fused at position 275 or 277 in uPAR, which eliminates the region of uPAR proximal to the GPI-attachment site (position 283) thereby eliminating the protein sequence that van Veen et al. propose to serve in exosite binding. It is possible that the citation of Cunningham et al. 2003 may be in error and what the authors actually used was a more recent strategy in which GFP is inserted between uPAR and the GPI-anchoring signal (Hellriegel et al. 2011, FASEB J 25, 2882-2897) leaving residues 276-283 of uPAR in close enough proximity to occupy a putative exosite in GDE3. Although potentially interesting, this suggestion is nevertheless also complicated by the fact that this region is completely different between mouse and human uPAR (and should be mirrored by a similar difference in GDE3). If not elaborated on experimentally this speculation should thus be omitted.*

These points are well taken. Yet we feel that some speculation is allowed in this case, also because the reviewer considers our suggestion ‘potentially interesting’. The uPAR-GFP and uPAR-FLAG constructs used in this study are those described in Caiolfa et al. JCB 2007 and Cunningham et al. 2003, respectively. In both of these constructs, the tag has been inserted within the disordered region between the membrane proximal domain 3 and the GPI anchoring signal. Although the exact insertion site of the tag within the disordered region is not the same in the two constructs, both insertions leave the mature uPAR protein intact and accessible for putative exosite interactions. Mapping the location of the exact exosite interaction and determining the selectivity of GDE3 towards uPAR-homologs goes beyond the scope the present paper, in our frank opinion.

It is true that GPI-anchor variants and the sequence of their pro-peptide could determine protein localization or selective cleavage by GPI-PLC’s. We now have obtained new data supporting the view that GDEs not only recognize PI-lipid moieties at the water-membrane interface (or the adjacent amino acids) but also the attached protein. We previously showed that GDE2 cleaves GPI-anchored glypican 6 (GPC6) (E. Matas-Rico et al. 2016). To show that GDE substrate specificity is not solely dependent on the GPI-anchor attachment or the unstructured region upstream of this signal, we constructed uPAR-GPC6 chimeras. We swapped the C-terminal region of uPAR with that of GPC6, and analyzed GDE2 enzymatic activity. GDE2 did not shed the uPAR-GPC6 chimaera, whereas GDE3 did (Van Veen et al., manuscript in preparation). Therefore, surface charge distributions could well determine protein-protein interactions and, consequently, selective substrate specificity of GDE3 versus GDE3.

*7) In the seventh paragraph of the Results and Discussion it states that invasion of pre-implantation embryos is a known uPA/uPAR-dependent process is incorrect. Although uPAR is expressed by giant trophoblasts, the fact that genetic ablation of uPAR has no effect on mouse fertility (or development, or hemostasis) suggests that this system is not essential for the invasion of pre-implantation mouse embryos (Bugge et al. 1995, JBC 270, 16886).*

Our original phrasing was a bit inaccurate. Although uPAR is not essential for embryonic development (neither is GDE3, for that matter; S. Sockanathan, personal communication), its absence may be overcome by redundant or compensatory mechanisms (Teesalu et al., 1998). Thus, uPAR and GDE3 appear to be ‘involved’ in the migration and invasion of pre-implantation embryos. We have revised the text accordingly, with new references.

*8) It would be interesting, and potentially important, to test the effects of GDE3 over-expression on the in vivo tumorigenicity and/or metastasis of MDA-MB-231 cells in appropriately xenografted mice. More specifically, to what extent does GDE3 over-expression diminish the in vivo tumorigenicity of MDA-MB-231 cells and to what extent can any GDE3-mediated inhibitory effects on tumorigenesis be reversed by the expression of the GDE3 resistant uPAR-TM construct?*

We agree that it would be interesting to test the effects of GDE3 over-expression in xenografted mice, although this would make revision of our manuscript within two months impossible. Nevertheless, we have performed the mice experiments. The new results show that GDE3 overexpression slows tumor growth (Figure 6). We did not the test the uPAR-TM construct in mice, because of time and costs considerations

[Editors' note: further revisions were requested prior to acceptance, as described below.]

*[…] 1. Results: "Flow cytometry analysis confirmed loss of uPAR from the plasma membrane by GDE3 (Figure 1), while TIRF microscopy revealed that uPAR loss occurred largely at the basolateral plasma membrane, where integrins normally mediate cell-matrix adhesion (Figure 1)."*

*"Loss" is a bit strong, and basolateral refers to a membrane domain that is larger than that seen by TIRF microscopy. Perhaps revise to:*

*"Flow cytometry analysis of GDE3 over-expressing cells confirmed decreased levels of uPAR in the plasma membrane (Figure 1), while TIRF microscopy revealed substantial loss of uPAR loss from the ventral surface (Figure 1)."*

We agree with the reviewers and have revised the text accordingly. Furthermore, we have included a picture showing the subcellular localization of GDE6 (Figure 1—figure supplement 1; previously “results not shown”), and describe its cDNA cloning in the Material and methods.

*2. Figure 1: Given the speculative nature of the model, please relabel the dashed yellow line as "proposed substrate interaction border".*

We have relabeled the yellow line as requested.

*3. Figure 4: upper images show confocal images (focal plane unclear, please clarify) and the lower images show super-resolution, which uses TIRF and thus only the basal membrane. The results show that over-expressed WT GDE3, but not catalytically dead GDE3, segregates away from the uPAR signal (lack of co-localization). Please provide quantification of co-localization over multiple cells. In addition, if the conclusions are based on the TIRF and not confocal data, then the conclusion should be modified to make it clear that GDE3 may induce uPAR relocalization rather than (or as well as) cleavage, because only the basal membrane was visualized.*

In HEK-uPAR2 cells, GDE3 expression results in a moderate decrease of total membrane-anchored uPAR (shown by FACS). When plated on vitronectin, however, both HEK-uPAR and MDA-MB-231 cells show a substantial localized loss of uPAR from the basal membrane. We have quantified uPAR patches on the plasma membrane of wild-type, GDE3-expressing and GDE3 knockout cells using confocal microscopy (ventral membrane plane). In the revised manuscript, we have done additional experiments to allow for adequate quantification of GDE3 and GDE3(H229A) co-localization with those uPAR patches (new panel in Figure 4, showing the Mander’s coefficient). We think it unlikely that GDE3 activity could induce uPAR ‘relocalization’, but we have weakened our conclusion.

*4. Figure 4: Please clarify how the experiments were carried out. In particular, because uPAR may affect ECM composition and vice versa, what ECM were the cells plated on? Please clarify if n=3 is the number of independent experiments. Why wasn't suPAR release measured in these experiments? If suPAR release from MDA cells is as low as for HEK cells in Figure 1, then it would not be feasible to see a decrease in MDA cells deleted for GDE3, but if MDA cells normally release detectable suPAR then the media of the GCE3 knockout cells should be assayed for suPAR.*

We now explain that the cells were plated on vitronectin, and that N=3 is the number of independent experiments. Concerning suPAR measurements in GDE3-depleted cells, this is a valid question. The problem is that GDE3 expression in breast cancer cell lines is relatively low (Barretina et al., 2012). We therefore preferred to rely on sensitive morphological assays in GDE3 knockout cells, very similar to the protocol used in our studies on GDE2-mediated glypican release. In that case, relatively little GDE2-mediated glypican release accounts for a prominent phenotype (Matas-Rico et al., Cancer Cell. 2016;30:548). Of note, also in GDE3-expressing HEK-uPAR cells the most striking response was an altered phenotype when the cells plated were on VN. In conclusion, highly localized release of GPI-anchored proteins (by either GDE3 or GDE2) can result in dramatic phenotypic outcomes. We have emphasized this point in the text.

*5. Figure 7: This experiment shows a modest but statistically significant reduction in tumor volume due to GDE3 over-expression. It is unfortunate that catalytically-dead GDE3 or uPAR-TM were not used as controls but we appreciate the time constraints. In light of the limited controls, we agree with the cautious interpretation in the discussion. However, even more caution may be justified because IHC analyses of solid human cancers show that the majority of uPAR is actually not found on the cancer cells themselves (only a few cancers deviate from this) but on the activated tumor microenvironment - mostly comprising immune cells. Therefore, the reviewers recommend adding a further caveat that the full implications of GDE3 expression on uPAR-dependent tumor growth need to be explored in more detail.*

We thank the reviewers for these comments. We have rephrased the text as suggested.

6. Please clarify in the methods the exact structure of the uPAR-TM construct. The Sidenius lab made several such constructs with the tag in different positions.

The structure of the uPAR-TM construct used has been described by Cunningham et al (2003), as mentioned in the text. We now elaborate on this construct in the Material and methods.

*7. Figure 5: The anti-uPAR antibody detects two bands. Are both bands depicting uPAR proteins?*

We have used two different uPAR antibodies in our study. Consistent with literature data on uPAR, we often see two bands (sometimes a smear, depending on assay conditions). According to previous studies (Hoyer-Hansen et al. JBC 267:18224;1992), the upper band represents full-length uPAR, the lower band proteolytically cleaved uPAR(D2+D3) that lacks the D1 domain. We mention this in the legend, including the above reference.

*8. Figure 1—figure supplement 1: It is shown that GDE3 competes with exogenous PI-PLC to deplete uPAR. It is unclear if PI-PLC is not working at all if GDE3 is highly expressed or less efficient. An additional lane showing uPAR and GDE3-expressing cells without PI-PLC would clarify this point.*

We have added a panel showing blots of GDE3-expressing HEK-uPAR cells without PI-PLC treatment. Together, the results show that PI-PLC is working less efficiently upon GDE3 overexpression.